# How Well Did the Healthcare System Respond to the Healthcare Needs of Older People with and without Dementia during the COVID-19 Pandemic? The Perception of Healthcare Providers and Older People from the SI4CARE Project in the ADRION Region

**DOI:** 10.3390/geriatrics8010021

**Published:** 2023-02-01

**Authors:** Stella Fragkiadaki, Dionysia Kontaxopoulou, Evangelia Stanitsa, Efthalia Angelopoulou, Dimosthenis Pavlou, Darja Šemrov, Simon Colnar, Mitja Lustrek, Bojan Blažica, Inga Vučica, Roberta Matković, Katarina Vukojevic, Ana Jelicic, Pietro Hiram Guzzi, Vlatka Martinović, Amina Pekmez Medina, Guido Piccoli, Margherita Menon, Srdjan Kozetinac, Miodrag Miljković, Chrysanthi Kiskini, Themis Kokorotsikos, Vasiliki Zilidou, Ivan Radević, John Papatriantafyllou, Eleftherios Thireos, Agis Tsouros, Vlado Dimovski, Sokratis G. Papageorgiou

**Affiliations:** 11st Department of Neurology, Aiginition University Hospital, Vasilissis Sofias Street 72-74, 11528 Athens, Greece; 2School of Topography and Geoinformatics, University of West Attica, Ag. Spyridonos Str., 12243 Aigalew, Greece; 3Faculty of Civic and Geodetic Engineering, University of Ljubljana Jamova cesta 2, 1000 Ljubljana, Slovenia; 4School of Economics and Business, University of Ljubljana Kardeljeva ploščad 17, 1000 Ljubljana, Slovenia; 5Department of Intelligent Systems, Jožef Stefan Institute, Jamova cesta 39, 1000 Ljubljana, Slovenia; 6Computer Systems Department, Jožef Stefan Institute, Jamova cesta 39, 1000 Ljubljana, Slovenia; 7Department of Gerontology, Teaching Institute for Public Health of Split and Dalmatian County, Vukovarska 46, 21000 Split, Croatia; 8Department for Research Data Collecting and Analysis, Teaching Institute for Public Health of Split and Dalmatian County, Vukovarska 46, 21000 Split, Croatia; 9Department of Anatomy, Histology and Embryology, University of Split School of Medicine, Šoltanska ul. 2, 21000 Split, Croatia; 10Municipality of Miglierina, Street B. Telesio 88040, Italy and University of Catanzaro, viale Europa, 88100 Catanzaro, Italy; 11Faculty of Medicine, University Hospital Mostar, 88000 Mostar, Bosnia and Herzegovina; 12Health Insurance and Reinsurance Fund of Federation of Bosnia and Herzegovina, Trg Heroja 14, 71000 Sarajevo, Bosnia and Herzegovina; 13ALOT, SI4CARE-TEAM Street Cipro, 16, 25124 Brescia, Italy; 14Special Hospital Merkur, Cara Dusana 3, 36210 Vrnjaka Banja, Serbia; 15Department of European Union, Projects of Regional Development Fund of Central Macedonia, Vas. Olgas 198, 54 655, Thessaloniki, Greece; 16Lab of Medical Physics & Digital Innovation, and Thessaloniki Active & Healthy Ageing Living Lab, School of Medicine, Aristotle University of Thessaloniki, 54124 Thessaloniki, Greece; 17Faculty of Economics, University of Montenegro, 37 Bulevar Jovana Tomaševića, 81000 Podgorica, Montenegro; 18National Health System, Athens Medical Society, Meandrou 23, 115 28 Athens, Greece; 19Department of Global Health, Boston University School of Public Health, 715 Albany Street, Boston, MA 02118, USA

**Keywords:** healthcare system responses, COVID-19 pandemic, older people, cognitive impairment, dementia, caregivers, Adrion–Ionian region, telemedicine, healthcare challenges, accessibility

## Abstract

One major challenge during the COVID-19 pandemic was the limited accessibility to healthcare facilities, especially for the older population. The aim of the current study was the exploration of the extent to which the healthcare systems responded to the healthcare needs of the older people with or without cognitive impairment and their caregivers in the Adrion/Ionian region. Data were collected through e-questionnaires regarding the adequacy of the healthcare system and were anonymously administered to older individuals and stakeholder providers in the following countries: Slovenia, Italy (Calabria), Croatia, Bosnia and Herzegovina, Greece, Montenegro, and Serbia. Overall, 722 older people and 267 healthcare stakeholders participated in the study. During the COVID-19 pandemic, both healthcare stakeholders and the older population claimed that the healthcare needs of the older people and their caregivers increased dramatically in all countries, especially in Italy (Calabria), Croatia and BiH. According to our results, countries from the Adrion/Ionian regions faced significant challenges to adjust to the special needs of the older people during the COVID-19 pandemic, which was possibly due to limited accessibility opportunities to healthcare facilities. These results highlight the need for the development of alternative ways of providing medical assistance and supervision when in-person care is not possible.

## 1. Introduction

Increased life expectancy in European countries has led to a growing percentage of the aging population which in turn poses significant challenges to the healthcare system [1]. Furthermore, the global economic crisis and the austerity measures that had been implemented in EU countries over the last decade have significantly compromised the equality of healthcare accessibility. Finally, the recent health crisis imposed by the COVID-19 pandemic has significantly affected both people’s lives as well as the capacity of healthcare professionals to provide their services [2].

Aspects of healthcare accessibility involve the efficient use of healthcare services especially by more vulnerable groups such as older people. More specifically, those aspects include the effective communication with healthcare professionals and healthcare facilities, the dissemination of practical information to the public, as well as organizational issues such as the projected timeframe for booking an appointment [3]. The older people were one of the most affected groups of the population due to the higher health risks they faced during the COVID-19 and additional factors that may have increased their difficulty in accessing healthcare services. For example, the older people are at a higher risk for cognitive decline and dementia, since age is usually one crucial factor for the development of mental disorders [4]. Furthermore, people with low social support and functional independence faced even more difficulties in accessing healthcare facilities and services, prolonging the time frames for follow-up and long-term treatment support due to the increased demands of the pandemic on the healthcare system [5].

In order to counteract the risks of visiting healthcare professionals imposed by COVID-19, health facilities have been adjusted to provide more safety in terms of probable spread of the disease and/or have adopted the provision of remote services [6]. Furthermore, in-person visits for individuals with chronic illnesses were highly non-advisable and at times completely unavailable due to the government restrictions, the focus of the healthcare system on urgent conditions, the overall fear of potential exposure to the coronavirus in healthcare facilities and the increased availability of tele-health services in comparison to the pre-pandemic period. The introduction of telemedicine during the pandemic offered significant benefits in terms of long-term monitoring, better compliance with the treatment regime and better follow-up rates as well as increased patient satisfaction, who often suffered from additional adverse effects of the pandemic, such as increased stress as well as reduced physical activity and sleep [7].

Nonetheless, accessibility to healthcare was significantly compromised especially for older people, as indicated by a recent study that integrated data from the SHARE Corona Survey and the SHARE Wave 7 from 25 EU countries and Israel (N = 40,919), which explored the effect of the COVID-19 pandemic to European citizens over 50 years old. This study highlighted the limited availability of healthcare services during the initial outbreak especially for residents of urban areas, people with increased health problems and needs, and those with financial hardships [8]. On the other hand, the context of providing tele-health services especially to the older people has not been thoroughly examined, as healthcare specialists in geriatrics have adequately participated neither in the development of guidelines and healthcare policies nor in the allocation of human and physical resources [9].

In terms of patients with cognitive impairment/dementia, due to the usually non-urgent nature of their healthcare needs, they refrained from visiting outpatient units either because the outpatient clinics themselves were temporarily closed to avoid transmission or because healthcare specialists were occupied elsewhere [10]. Apart from the patients with cognitive impairment/dementia, caregivers were also significantly affected by the COVID-19 pandemic. According to a recent study [11], family caregivers were requested to outline how they could have been better supported during the pandemic. The participants responded that the pandemic highlighted the already existing difficulties and deficiencies in long-term care, especially in terms of human resources and management support. They stated that although the healthcare system’s response to the initial outbreak of the pandemic was as expected, the adjustment to their needs was much longer than originally anticipated. Furthermore, they also experienced an exclusion from medical evaluations and communication was lacking, especially in terms of sharing their knowledge regarding the patient. According to them, all aspects of long-term care such as home care services and supportive living were only partially addressed, while their own caregiver burden, stress and feelings of depression and loneliness were further increased during the COVID-19 pandemic. The only positive aspect that the respondents noted was the avoidance of transportations, since most of the medical appointments and support sessions were carried out online [11].

One of the main changes that the COVID-19 pandemic forced onto the healthcare systems globally was a rapid transition from in person visits to online medical appointments [12]. Although the use of tele-neurology is not a new practice, as it has been regularly used in the medical care of acute stroke patients, it had not been widely utilized by healthcare providers for other neurological conditions. The increased need for online medical support accelerated the shift of healthcare facilities and healthcare providers for patients with chronic neurological disorders. Furthermore, public authorities were also forced to address the legislative aspects of tele-care, provide directives, revise already existing guidelines to facilitate healthcare providers and ensure the protection of personal data of the patients [13,14,15].

Although during the COVID-19 pandemic some adjustments were made to the healthcare system in order to accommodate the needs of the older people, it seems that a significant proportion of their needs remained unaddressed. According to the current literature, the COVID-19 pandemic significantly affected the provision of healthcare services globally. The results of the available research suggest that the vulnerable group of the older people, and even more of the older people with cognitive impairment and dementia were significantly affected from the effects of the COVID-19 pandemic, especially in terms of accessibility to healthcare services and in terms of their overall functionality and their ability to perform everyday activities without restrictions. A significant inadequacy of previous research is the lack of consistency among reported methodologies and the sample selected. Additionally, a significant proportion of them were literature reviews and were not based on empirical data, while there is inadequate evidence regarding the perspectives of healthcare providers and healthcare services users in a single study which could provide a better overview of the existing problems of the healthcare system. To the best of our knowledge, there is no study that explores the views of stakeholders and healthcare receivers regarding patients with cognitive impairment/dementia and whether this specific group faced more challenges during the pandemic in comparison to their cognitively healthy counterparts. In addition, relative evidence in Adriatic–Ionian regions is lacking, and the potential differences between these countries remain unknown. The aim of the current study was to explore how the healthcare system in Adriatic–Ionian countries/regions responded considering the needs of older people with or without cognitive impairment during the COVID-19 pandemic, according to the opinions of the older people and healthcare providers. Based on the observations of the existing literature, and the overall effect of COVID-19 pandemic on the healthcare systems, it was hypothesized that the accessibility of the older population with or without cognitive impairment to healthcare professionals would be limited and inadequate.

The current study is part of the SI4CARE project, which is supported by the Interreg ADRION Program, funded under the European Regional Development Fund and IPA II fund. The project’s main objective was to contribute to the creation of a transnational effective ecosystem for the social innovation application in integrated healthcare services for the ageing population across ADRION countries through a joint collaboration network and a unique strategy translated into regional and national action plans, implemented and monitored within pilots, once innovative approaches have been tested and backboned by an ICT decision support system. This objective contributes to the topic social innovation as SI4CARE aims to tackle the needs of the ADRION ageing population for long-term healthcare by creating a collaborative environment where it is important to co-design solutions and engage a large transnational community.

## 2. Materials and Methods

### 2.1. Sample

The sample of the present study included two target groups: (a) healthcare providers (later referred to as stakeholders—SH), people who provide health services to patients and maintain health information about them, derived from various positions in healthcare services for the older people, for example: non-governmental organization (NGOs), geriatric societies, medical societies, universities, regulatory authorities, social innovator experts, care centers for the older people, and healthcare service providers and (b) general older persons above the age of 65.

The total number of the interviewees was 722 older people and 267 healthcare stakeholders in all regions. More specifically: (a) 124 older people and 39 healthcare stakeholders from Slovenia, (b) 111 older people and 35 healthcare stakeholders representing Calabria–Italy, (c) 96 older people and 31 healthcare stakeholders from Croatia, (d) 100 older people and 30 healthcare stakeholders from Bosnia and Herzegovina, (e) 88 older people and 57 healthcare stakeholders, representing Greece, (f) 103 older people and 45 healthcare stakeholders from Montenegro and (g) 100 older people and 30 healthcare stakeholders from Serbia. 

### 2.2. Procedure 

Data were collected through e-questionnaires regarding the adequacy of the healthcare system to the needs of the older people with or without cognitive disorders and were anonymously administered to older people and stakeholder providers in the following countries: Slovenia, Italy (Calabria region), Croatia, Bosnia and Herzegovina, Greece and Serbia. It should be noted that with the exception of Italy where the questionnaires were administered only to the Calabria region, all the other participating countries collected data at a national level.

For the development of the questionnaire and the selection of the appropriate questions, desk research was conducted for identifying similar questionnaires previously administered to the older people and/or relevant stakeholders regarding the healthcare systems in Europe. Furthermore, for the finalization of the questionnaire items, two experienced Greek healthcare specialists, who have been closely involved in matters of public health through the World Health Organization (WHO) (AT) and National Health Associations (ET), provided advice. Consequently, a pilot sample was selected (both stakeholders and older people) for the administration of the questionnaires to assess if the information required was clearly explained and if they could be easily completed by an older individual independently of their educational status.

The questionnaire involved two subscales: (a) a 29-item subscale regarding the healthcare system for the older people and (b) a 23-item similar subscale regarding the healthcare system for individuals with cognitive disorders/dementia. The answers of the questions were structured in a 5-point Likert scale (0–4: 0—not at all, 1—slightly, 2—moderately, 3—very, 4—extremely). The questions were divided in five separate topics: (a) Availability, (b) Affordability, (c) Accessibility, (d) Adequacy and (e) Appropriateness according to the WHO elements for accessibility in healthcare systems. The questionnaire also contained relevant demographic data, including sex, year of birth, country and region of residency, and education level. 

For this study, from the overall questionnaire, the questions related to how the health system responded during the COVID-19 pandemic were selected for the analyses. More specifically, the questions used for the current study were: (1) “To what extent did the healthcare system respond to the healthcare needs of the older people during the COVID-19 pandemic?”, (2) “Compared to the younger ones, how much more difficult was the daily life of the older people during the COVID-19 pandemic?”, (3) “During the COVID-19 pandemic, to what extent did the healthcare needs of the older people with memory impairment/dementia increase?”, (4) “During the COVID-19 pandemic, to what extent did the healthcare system respond to the healthcare needs of the older people with memory impairment/dementia?”, (5) “During the COVID-19 pandemic, to what extent did the healthcare system respond to the healthcare needs of the caregivers for the older people with memory impairment/dementia?”. It should be noted that Slovenia did not provide data for the questionnaires regarding patients with cognitive disorders/dementia. Thus, only data for the general older people population from Slovenia were included in the analysis.

### 2.3. Ethics

Prior to the administration of the questionnaires, all participants had been informed of the purpose of the SI4CARE project and of the procedure of their participation. For the administration of the questionnaires, all participants needed to provide consent for their participation in the survey. It was highlighted to them that their participation was voluntarily, and they were reassured that the information obtained during the completion of the questionnaires would be used only for the purpose of the above-mentioned survey. As such, confidentiality and informed consent were ensured for all participants in order to adhere with the ethical guidelines that ensure good practice procedures in the conduction of empirical research.

### 2.4. Statistical Analysis

Absolute frequencies were calculated for the demographic characteristics of the sample, separately for the SHs and the older people. GLM mixed two-way ANOVAs were used with country of residence (Italy, Croatia, BiH, Serbia, Greece, Montenegro) as between-subject variable and type of group (healthy older people vs. older people with memory impairment/dementia) as within-subject variables in order to assess their possible effects on the perceptions of the SHs and the older people. One-way ANOVAs were used to explore the possible effects of country of residence to the remaining questions related to COVID-19. Analyses were conducted using IBM SPSS Statistics v22.0, and the statistical significance level was *p* < 0.05.

## 3. Results

Demographic characteristics of the SHs and the older participants from each country are presented in Table 1.

### 3.1. Perceived Response of the Healthcare System by the SHs

Mixed two-way ANOVA showed that opinions of SHs’ (about healthy older people vs. older people with cognitive disorders/dementia) had a statistically significant main effect on their perception of the extent the healthcare system of their country of residence responded to the healthcare needs of the older people during the COVID-19 pandemic (*F*(1,215) = 8.018, *p* = 0.005, η^2^_p_ = 0.04). SHs reported a significantly lower perceived response of the healthcare system for older people with cognitive disorders dementia compared to the healthy older people. There was also a significant main effect of country of residence (*F*(1,215) = 10.71*, p* < 0.001*,* η^2^_p_ = 0.20). SHs in Italy reported a significantly lower perceived response of the healthcare system compared to all the other countries, while in Croatia, BiH, Serbia, Greece and Montenegro, there were no significant differences. The interaction between the opinions and country of residence was not significant (*F*(5,215) = 0.81, *p* = 0.543). The descriptive statistics of the perceived responses of the SHs are presented in Table 2.

### 3.2. Perceived Response of the Healthcare System by the Older People

Mixed two-way ANOVA showed that the opinions of the older people (about healthy older people vs. older people with cognitive disorders/dementia) had a statistically significant main effect on their perception of the extent the healthcare system of their country of residence responded to the healthcare needs of the older people during the COVID-19 pandemic (*F*(1,450) = 34.23, *p* < 0.0001, η^2^_p_ = 0.07). The older people reported a significantly lower perceived response of the healthcare system for the older people with cognitive disorders/dementia compared to the healthy older people. There was also a significant main effect of country of residence (*F*(1,450) = 30.90, *p* < 0.001*,* η^2^_p_ = 0.26). Older people in Italy reported a significantly higher perceived response of the healthcare system, compared to Croatia and BiH, and significantly lower perceived response of the healthcare system compared to Serbia. Older people in Croatia reported a significantly lower perceived response compared to Italy and Serbia. Older people in BiH reported a significantly lower perceived response of the healthcare system compared to Italy, Serbia and Montenegro. Older people in Serbia reported a significantly higher perceived response of the healthcare system compared to Italy, Croatia, BiH and Greece. Older people in Greece reported a significantly lower perceived response of the healthcare system compared to Serbia, but there were no differences with other countries. Older people in Montenegro reported a significantly higher perceived response of the healthcare system compared to Croatia and BiH. The descriptive statistics of the perceived responses of the older people are presented in Table 2.

The interaction between the opinions and country of residence was significant (*F*(1,450) = 6.53 *p* < 0.001, η^2^_p_ = 0.07). Country of residence had a significant effect on perceived response of the healthcare system to the needs of older people (*F*(5,450) = 27.10 *p* < 0.001, η^2^_p_ = 0.23) and on the perceived response of the healthcare system to the needs of older people with memory impairment/dementia (*F*(5,450) = 20.13 *p* < 0.001, η^2^_p_ = 0.18) (Figure 1).

### 3.3. Perceived Difficulties of the Older People by the SHs and the Older People

One-way ANOVAs were used to examine the possible effects of country of residence to the perceived difficulties of the older people compared to the younger individuals during the COVID-19 pandemic. Country of residence had an important effect both for the SHs (*F*(6,249) = 26.36, *p* < 0.001, η^2^ = 0.39) and the older people (*F*(6,613) = 3.63, *p* = 0.002, η^2^ = 0.03), as reported at Table 3.

### 3.4. Perceived Increase in the Healthcare Needs of the Older People by the SHs and the Older People

One-way ANOVAs were used to examine the possible effects of country of residence to the perceived increase in the healthcare needs of the older people with memory impairment/dementia during the COVID-19 pandemic. Country of residence had an important effect both for the SHs (*F*(1,193) = 18.06, *p* < 0.001, η^2^ = 0.32), and the older people (*F*(5,425) = 7.75, *p* < 0.001, η^2^ = 0.08), as reported at Table 3. 

### 3.5. Perceived Response of the Healthcare System to Caregivers by the SHs and the Older People

One-way ANOVAs were used to examine the possible effects of country of residence to the perceived response of the healthcare systems to the healthcare needs of the caregivers for the older people with memory impairment/dementia during the COVID-19 pandemic. Country of residence had an important effect both for the SHs (*F*(1,193) = 10.21, *p* < 0.001, η^2^ = 0.21) and the older people (*F*(1,428) = 20.77, *p* < 0.001, η^2^ = 0.20) as reported at Table 3.

## 4. Discussion

The aim of the current study was to explore for the first time the perspectives of the older people and healthcare stakeholders on how well the healthcare system responded to the healthcare needs of the older people with and without cognitive impairment and their caregivers during the COVID-19 pandemic in Adriatic–Ionian regions. Results showed that the healthcare systems responded poorly to the needs of the older people during the outbreak of the pandemic, while older people with cognitive disorders/dementia and their caregivers were even more significantly affected by the above outcome. Considering the country of residence, SHs in Italy (Calabria region) reported a significantly lower perceived response of the healthcare system compared to all the other countries, while the older people in Italy (Calabria region) reported a significantly higher perceived response of the healthcare system compared to Croatia and BiH and significantly lower compared to Serbia. Older people in BiH reported a significantly lower perceived response of the healthcare system compared to Italy (Calabria region), Serbia and Montenegro. In addition, country of residence had an important effect on the perceived difficulties of the older people by the SHs and the older people, the perceived increase in the healthcare needs of the older people, as well as for the perceived response of the healthcare system to caregivers.

According to the responses from both SHs and older participants, older people with cognitive impairments/dementia and their caregivers faced more difficulties receiving healthcare services in comparison to older people without cognitive disorders. This result was in accordance with the existing literature describing the inadequate responsiveness of the healthcare system to patients with non-urgent medical conditions, such as cognitive disorders and dementia [10,16]. During the peak of the pandemic, in-person medical visits were strictly prohibited or avoided, and other alternatives, such as telemedicine, had not yet been adequately developed. Furthermore, the limited capacity of healthcare professionals during the pandemic did not allow anyone to prioritize the management of patients with chronic diseases, in contrast to those with more acute medical conditions. Under those circumstances, patients could carry on with their usual treatment regime and medication but any other need, such as changes in their clinical status that needed to be addressed or scheduled clinical assessments (blood tests, neuroimaging appointments) were not adequately met. Furthermore, the initial diagnosis of patients with cognitive impairment was also delayed due to lack of available appointments, prolonging the initiation of appropriate management, thus enhancing the burden of the caregiver and the distress of the patient [16]. Moreover, holistic management of patients with cognitive impairment/dementia, including physical exercise, social interaction and engagement, participation in activities of day care centers and rehabilitation programs was also refrained, which also negatively affected the clinical and especially the psychological condition of the patients and their caregivers [16,17,18].

Regarding the caregivers, previous studies support that the caregivers of older people with cognitive decline or dementia faced many difficulties during the COVID-19 pandemic. For example, qualitative interviews were carried out with patients with cognitive impairment/dementia and their caregivers during the initial phase of the pandemic in England. According to their responses, they felt comforted by check-up calls provided by healthcare professionals, but they also consciously avoided receiving healthcare services, to avoid infection, to minimize the burden of the National Health System, or due to their lack of understanding of technological equipment. Furthermore, remote medical evaluations introduced some additional hardships to the communication between the patient and the healthcare professional, including missing and rescheduling calls, lack of indications to remember problems as well as difficulties engaging the patient with cognitive impairment/dementia [19]. In another study, caregivers often reported feelings of isolation, fear and stress, lack of proximity with family members, inability to travel and inability to walk around local markets. The additional confinement at home with an older individual with cognitive and sometimes psychiatric or behavioral problems further enhanced the feeling of helplessness and depression [16]. From another survey that was carried out online in Italy and Hungary, which involved caregivers of demented patients during the initial wave of the COVID-19 pandemic, it was reported that a substantial percentage of the participants suffered from a significant deterioration of their financial situation along with impaired mental and physical health due to lack of proper support structures. The results of the survey highlighted the need to focus on the care of the caregivers along with the patients with dementia and re-design the healthcare system to accommodate for their own needs as well [5].

Interestingly, significant differences were observed according to the country of residence from the respondents. It should be noted that countries from Adriatic–Ionian regions that were included in the current study have complex health systems divided into primary, secondary and tertiary care, while most of them utilize public and private services. Thus, it comes as no surprise that the healthcare system on all countries and regions faced significant challenges adjusting to the urgent needs that arose during the COVID-19 pandemic while still putting an effort on continuing providing healthcare services to those in need. Thus, healthcare providers seem to have envisioned changes and adaptations in order to meet the increased demands, but the needs still could not be met. The older people had the greatest difficulties as due to the pandemic, their access to health structures was more difficult. Some countries tried to adjust to these difficulties by adopting distance examination methods using technology, such as telemedicine or consultation through the telephone. Overall, the COVID-19 pandemic seemed to highlight the need to find reflexive methods of assessment and access to the health system.

In our study, Italy (Calabria region) was one of the participating countries with the most diverse opinions between the stakeholders and the older people in regard to the responsiveness of the healthcare system during the COVID-19 pandemic, where Italian stakeholders reported significantly worse response to the healthcare system in comparison to the older people. This result could reflect the high regional nature of the healthcare system in Italy and the specific characteristics of the Calabria region. It could be possible that healthcare providers in the Calabria region are more affected from the decentralized nature of the healthcare systems and the complexity that this could involve and the difficulty that this would impose on quickly adjusting to urgent medical needs [20].

Croatia was also a country that reported significant challenges, especially regarding the perceived response of the healthcare system to caregivers of patients with cognitive impairment/dementia during the COVID-19 pandemic. This could be a result, as until recently, of the lack of initiatives regarding the management of patients with dementia as well as the lack of official protocols and action plans for the long-term treatment of dementia patients [21]. Bosnia and Herzegovina also reported significant challenges in terms of the perceived response of the healthcare system to the needs of caregivers during the COVID-19 pandemic. This could also be a result of the organization of the healthcare system in BiH, a significant proportion of which is decentralized and distributed to ten cantonal ministries of health, resulting in significant friction when urgent adjustments need to be made. Furthermore, no official guidelines and procedures are available further enhancing the perceived difficulties that were reported for the response of the healthcare system to the caregivers during the COVID-19 pandemic [22].

On the other hand, Serbia and Montenegro were two of the participating countries where the responses both from the SHs as well as from the older people qualitatively were around the middle values, indicating that no significant perceived changes at the provision of healthcare services were observed during the COVID-19 pandemic, especially for the older people population without cognitive disorders/dementia. This is an interesting finding, since both countries also faced significant challenges and difficulties during the COVID-19 pandemic [23,24]. 

According to our results, countries from the Adrion regions faced significant challenges to adjust to the special needs of older people with cognitive impairment and their caregivers during COVID-19 pandemic, which was possibly due to accessibility opportunities to healthcare facilities. These results highlight the need for the development of alternative ways of providing medical assistance and supervision when in-person care is not possible. The recent crisis in global health through the spread of the COVID-19 pandemic promoted the utilization of telemedicine as an effective tool of providing healthcare services on time and at the same time eliminating the risk of probable infection [25]. Such a development has proved particularly useful for the older people population, whose health and well-being were at increased risk during the pandemic [26]. In a study interviewing caregivers of people with dementia, some of them reported the importance of covering immediate needs, and others highlighted the importance of long-term needs during the pandemic and suggested methods to compensate for those needs, such as using tele-consultation [27]. In addition, distance health services have been developed in Italy for people with dementia and for providing support to caregivers [28]. Furthermore, in Spain, telemedicine was found to be inadequate in terms of supporting patients with mild cognitive impairment/dementia who were living at home [29]. 

The current study had some limitations that should be reported. The methodological design of the study is compatible with a survey design, meaning that all the reported answers, both by the older people and the SHs are their perceptions for the healthcare system in their countries of residence. In addition, for Italy, data were only collected from the region of Calabria, while other countries were represented by collection of data at a national level. Although gender differences were present in the current study, they were not further analyzed, as the scope of the survey was the opinion of SHs and older people. Along this vein, previous studies had not included gender as a significant variable that would affect their overall opinion regarding the healthcare system. Moreover, the e-questionnaires used for the current study were not validated but were designed based on the previous related literature and the opinion of public health experts, as no related tool to address the issues was available in the current study. Finally, the available literature regarding the healthcare responses for each country/region in the Adrion/Ionian territory was limited in order to provide a cohesive discussion of the observed results. This observation highlights the need for further investigation on the healthcare responses for different countries. 

In conclusion, the current study highlighted the existing accessibility issues that arose during the COVID-19 pandemic as well as the need for the re-organization of the healthcare systems in order to facilitate the development and utilization of sustainable interventions that would not require the physical presence of the patient, which could serve as a future opportunity to simplify the management of long-term care to ensure the availability of continuous care to those in need [30]. Future studies could explore this issue at a transnational level with an ultimate goal of the development of a common strategy and relevant action plans among countries and regions, which will be based on current advances in technology. 

## Figures and Tables

**Figure 1 geriatrics-08-00021-f001:**
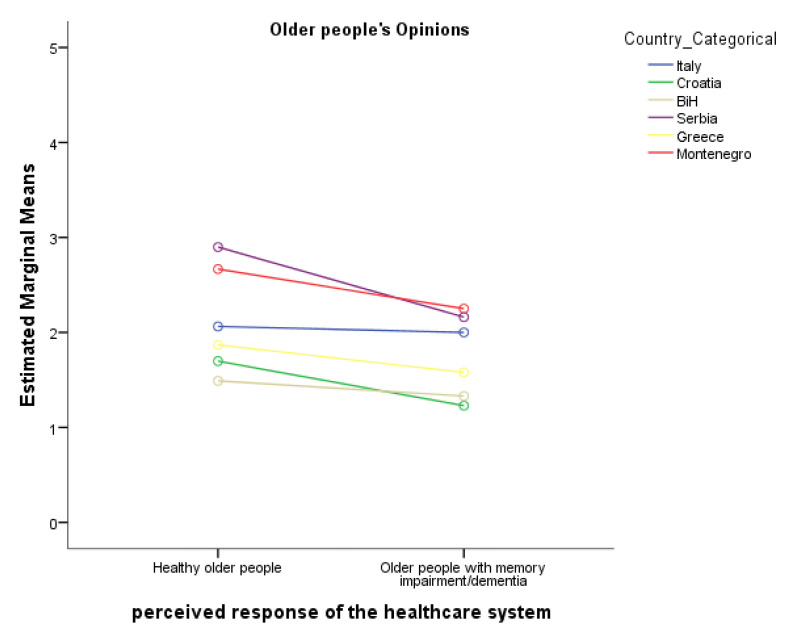
Simple effects from the interaction between the opinions of perceived response of the healthcare system and the country of residence.

**Table 1 geriatrics-08-00021-t001:** Demographic characteristics of SHs and older participants.

	SHs (n = 267)	Older People (n = 722)
Demographic Information	*f*	*f*
**Gender**		
Men(women)	91(176)	279(443)
**Country**		
Slovenia	39	124
Italy	35	111
Croatia	31	96
BiH	30	100
Serbia	30	100
Greece	57	88
Montenegro	45	12
**Educational Level**		
Incomplete primary	-	24
Primary	-	189
Secondary	54	264
BSc	139	143
MSc	54	71
PhD	17	21

Note: BiH: Bosnia–Herzegovina, BSc: Bachelor’s Degree, MSc: Master’s Degree, PhD: Doctor of Philosophy.

**Table 2 geriatrics-08-00021-t002:** Descriptive statistics of perceived response of the healthcare system during the COVID 19 pandemic by older people group (with vs. without cognitive impairment) and country of residence.

	Older People without Cognitive Impairment		Older People with Cognitive Impairment	
Perceived Response	SHs M (±*SD*)	Older People M (±*SD*)	SHs M (±*SD*)	Older People M (±*SD*)
Italy	0.74 (±0.74)	2.06 (±0.79)	0.77 (±0.65)	2.00 (±0.00)
Croatia	2.00 (±1.51)	1.70 (±1.04)	1.65 (±1.33)	1.23 (±0.96)
BiH	1.80 (±1.09)	1.49 (±0.99)	1.63 (±1.10)	1.33 (±0.89)
Serbia	2.07 (±1.26)	2.90 (±0.94)	2.07 (±1.31)	2.16 (±1.08)
Greece	2.16 (±0.98)	1.87 (±1.06)	1.92 (±1.02)	1.58 (±0.89)
Montenegro	2.36 (±1.00)	2.67 (±0.49)	2.09 (±0.95)	2.25 (±0.62)

Note: SHs: stakeholders, BiH: Bosnia–Herzegovina.

**Table 3 geriatrics-08-00021-t003:** Descriptive statistics of perceived response of the healthcare system during the COVID-19 pandemic to caregivers and country of residence.

		*F*	*p*	η^2^	Post Hoc
Perceived difficulties older people vs. younger individuals	SHs	26.36	<0.001	0.39	IT < SL **, IT < CR **, IT < SE **, IT < BiH **, IT < GR **, IT < MoN **
	Older people	3.63	0.002	0.03	IT > CR **, IT < SE **, IT > BiH **, IT < GR **, IT > MoN **
Perceived increase in the healthcare needs of the older people with memory impairment/dementia	SHs	18.06	<0.001	0.32	IT < CR **, IT < SE **, IT < BiH **, IT < GR **, IT < MoN **
	Older people	8.09	<0.001	0.08	IT < CR **, IT < SE **, IT < BiH *, IT < MoN *
Perceived response of the healthcare systems to caregivers for the older people with memory impairment/dementia	SHs	10.21	<0.001	0.21	IT < CR **, IT < BiH **, IT < SE *, IT < GR *, IT < MoN *
	Older people	20.77	*p* < 0.001	0.20	IT < SE **, IT < MoN **, CR < SE **, CR < MoN **, BiH < SE **, BiH < MoN **

Note: SHs: stakeholders, SL: Slovenia, ΙΤ: Italy, CR: Croatia, SE: Serbia, MoN: Montenegro, BiH: Bosnia–Herzegovina, * *p* < 0.05, ** *p* < 0.001.

## Data Availability

Data available on request due to restrictions e.g., privacy or ethical.

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
