# Peer review of "How Well Did the Healthcare System Respond to the Healthcare Needs of Older People with and without Dementia during the COVID-19 Pandemic? The Perception of Healthcare Providers and Older People from the SI4CARE Project in the ADRION Region"

_geriatrics, 2023, doi:10.3390/geriatrics8010021_

Round 1

Reviewer 1 Report

This paper is an empirical study. In that case, you should clearly state the hypothesis you want to verify by research in the first half of the paper, and discuss it in the second half.

Only brief results are given for country-specific results. You should discuss what you can read from the results by referring to previous studies.

A small number of references. By referring to more literature, (1) how far the previous research has reached, (2) what problems existed in the previous research, and (3) what improvements this research has made, must be clearly stated.

Author Response

Point 1: This paper is an empirical study. In that case, you should clearly state the hypothesis you want to verify by research in the first half of the paper, and discuss it in the second half. 

Response 1: Thank you for your comment. We have added a hypothesis to the introduction section as suggested. We would like to note that, based on the literature, we would expect that their opinions would align with restricted access to healthcare services, both for cognitively healthy and cognitively impaired individuals. However, not enough data were availabe in order to conclude on a certain hypothesis.

Point 2: Only brief results are given for country-specific results. You should discuss what you can read from the results by referring to previous studies.

Response 2: Thank you for your comment. We understand that the discussion for each country was limited. However, we should point out that the existing literature was not also very limited and also linguistic restricitions were often rised. However, this is an important point to address, so we have added it as an additional limitation to the final paragraph of the manuscript.

Point 3: A small number of references. By referring to more literature, (1) how far the previous research has reached, (2) what problems existed in the previous research, and (3) what improvements this research has made, must be clearly stated.

Response 3: Thank for you comment. We have further recised the introduction section in order to address clearly your comment, which we believe is important to note. Unfortunately, as stated previously, the existing literature is limited and made difficult the extraction of safe cocnlusions apart from the very obvious ones. According to our own revision of the literature, we noticed that the available studies were not consistent in terms of methodology and sample selected and a significant proportion of them were literature reviews and were not based on empirical data. Finally, no other studies combine on the same research the opinions of both SHs and the affected populations in countries of the Adrion / Ionian region.

Reviewer 2 Report

I would like to thank the authors for their work.

This is an interesting paper, which aims to explore how the healthcare system in Adriatic – Ionian countries / regions responded considering the needs of elderly with or without cognitive impairment during the COVID-19  pandemic, according to the opinions of the elderly and healthcare providers.

The abstract is clear, the introduction well-deepened and up to date.

The adopted method is consistent with the aim of the study. I would suggest to add in the ethics section whether the acquisition of the opinion of the ethics committee was necessary; alternatively, I would specify that it was not necessary to refer to it.

The results are clear, the discussion is consistent with them and the background. The findings of this study may be used as a first step to identify the criticism of the different healthcare services and to exchange good practices among the involved countries.

Author Response

Point 1: The adopted method is consistent with the aim of the study. I would suggest to add in the ethics section whether the acquisition of the opinion of the ethics committee was necessary; alternatively, I would specify that it was not necessary to refer to it.

Response 1: We would like to thank the reviewer for their supportive comments on the overall content of the manuscript.

Regarding the ethics section, added to the manuscript and additional sentence to address this comment (l. 230-232)

Reviewer 3 Report

The study analyses the extent to which the healthcare systems responded to the healthcare needs of the elderly with or without cognitive impairment and their caregivers in different countries in the Adrion/Ionian region. The manuscript is clear and relevant and significant for the field of the health system in covid 19. Present a satisfactory revision of the literature and a description of the procedures. The conclusions are consistent with the evidence and improve the knowledge for future interventions in this field, special in healthcare systems for elderly people. 

English language and style are fine/minor spell check required; some writing errors are found in the text (e.g. line 69 “om mental disorders”) and it is important coherence in the terms (e.g. COVID- 19 pandemic or COVID-19 epidemic line 376)

The graphic should be changed as the word is misspelled "impariment"

Table 3 - Revise the third item to capitalize like the others

Some references are not formatted according to the rules of the journal and must be reviewed (e.g. line 102, line 414, line 416)

It is suggested made a minor revision of the writing style

Author Response

Response to Reviewer: Thank you for your comments. The manuscript has been carefully revised to correct any typographic and formatting errors.

Round 2

Reviewer 1 Report

The author should discuss his hypothesis in the later part of the article. In other words, he should state whether he can say that the analysis in this paper supports his hypothesis.